# Elucidating Recombination Mediator Function Using Biophysical Tools

**DOI:** 10.3390/biology10040288

**Published:** 2021-04-01

**Authors:** Camille Henry, Sarah S. Henrikus

**Affiliations:** 1Department of Biochemistry, University of Wisconsin-Madison, Madison, WI 53706, USA; 2Macromolecular Machines Laboratory, The Francis Crick Institute, London NW1 1AT, UK

**Keywords:** recombination mediator protein, single-strand binding protein, recombinase, recombination, DNA repair, biophysical tool

## Abstract

**Simple Summary:**

This review recapitulates the initial knowledge acquired with genetics and biochemical experiments on Recombination mediator proteins in different domains of life. We further address how recent *in vivo* and *in vitro* biophysical tools were critical to deepen the understanding of RMPs molecular mechanisms in DNA and replication repair, and unveiled unexpected features. For instance, in bacteria, genetic and biochemical studies suggest a close proximity and coordination of action of the RecF, RecR and RecO proteins in order to ensure their RMP function, which is to overcome the single-strand binding protein (SSB) and facilitate the loading of the recombinase RecA onto ssDNA. In contrary to this expectation, using single-molecule fluorescent imaging in living cells, we showed recently that RecO and RecF do not colocalize and moreover harbor different spatiotemporal behavior relative to the replication machinery, suggesting distinct functions. Finally, we address how new biophysics tools could be used to answer outstanding questions about RMP function.

**Abstract:**

The recombination mediator proteins (RMPs) are ubiquitous and play a crucial role in genome stability. RMPs facilitate the loading of recombinases like RecA onto single-stranded (ss) DNA coated by single-strand binding proteins like SSB. Despite sharing a common function, RMPs are the products of a convergent evolution and differ in (1) structure, (2) interaction partners and (3) molecular mechanisms. The RMP function is usually realized by a single protein in bacteriophages and eukaryotes, respectively UvsY or Orf, and RAD52 or BRCA2, while in bacteria three proteins RecF, RecO and RecR act cooperatively to displace SSB and load RecA onto a ssDNA region. Proteins working alongside to the RMPs in homologous recombination and DNA repair notably belongs to the RAD52 epistasis group in eukaryote and the RecF epistasis group in bacteria. Although RMPs have been studied for several decades, molecular mechanisms at the single-cell level are still not fully understood. Here, we summarize the current knowledge acquired on RMPs and review the crucial role of biophysical tools to investigate molecular mechanisms at the single-cell level in the physiological context.

## 1. Discovery and Initial Phenotypes Observed for RMPs

This chapter summarizes genetic studies which identified and helped to characterize the function of Recombination Mediators Proteins (RMPs) as well as their partners in DNA replication and repair across different kingdoms of life.

### 1.1. Discovery and Phenotype of Bacterial RMPs—RecFOR 

In bacteria, RMPs are encoded by *recF*, *recO,* and *recR* genes which belong to the same DNA repair pathway (Table 1). Their discovery is the product of different studies. Horii and Clark were the first to use an *Escherichia coli* deletion mutant, encoding RecB, RecC, SbcB, and SbcC nucleases, thus allowed them to identify additional genes involved in homologous recombination (HR), including *recF* [1]. This genetic approach paved the way to identify other members of this pathway, eventually leading to the discovery of *recO* and *recR* [2,3]. Due to their phenotypic resemblances, *recF*, *recO,* and *recR* were grouped with other genes (*recJ*, *recQ*, *recN*, …) in the *recF* epistatic group. Genetic experiments revealing the similarity between SSB overexpression and *recF* deletion were critical to understand the role of the RecF pathway in SSB removal [4,5]. *recFOR* deletion mutants are generally characterized by functional defects related to RecA presynaptic nucleofilament (called RecA*) activities, such as: (1) conjugation defect, (2) UV, Gamma rays and drug sensitivity, (3) delayed SOS induction, (4) translesion synthesis (TLS) defects and (5) replication fork instability [1,2,3,6,7,8,9,10,11]. Altogether, these data support a mechanism in which the three bacterial RMPs (RecF, RecO, and RecR) work in the same homologous recombination repair pathway [12,13], mediated by ssDNA (also called gap repair). Interestingly, in bacteria such as *Deinococcus radiodurans* naturally lacking the *recBC* genes, the RecF pathway plays a significant role in double strand break (DSB) repair through extended synthesis-dependent strand annealing process [14]. 

Phylogenetic analysis reveals considerable variability in *recF*, *recO,* and *recR* gene conservation despite their apparent epistasis [15]. In particular, *recO* or a *recO* ortholog gene is present in most bacterial species, whereas the *recF* gene is much less widely distributed [15,16]. This suggest either a redundancy of other DNA repair genes or a more complex mechanism in which RMPs have distinct functions but often cooperate with each other. Consistent with the second possibility, a rising number of studies revealed phenotypic differences between *recO* and *recF* in *Bacillus subtilis* and *E. coli* [17,18,19,20,21]. 

### 1.2. Discovery and Phenotype of the Bacteriophages RMPs—UvsY/Orf

In bacteriophage T4 (lytic), identification of DNA recombination genes was carried out using genetic screening of deletion mutants presenting sensitivity to UV, DNA damaging agents, such as ethyl and methyl methanesulfonate (EMS and MMS), hydroxyurea (HU), and furthermore showed HR defects [22,23,24,25,26,27]. First, *uvsX* was identified, then *uvsY,* and finally *uvsW* [23,24,26]. Sedimentation, phage plaque, and burst size analysis of single-deletion mutants revealed defects in DNA compaction [23,26,28]. Despite harboring some phenotypic differences, it was established that *uvsX*, *uvsW,* and *uvsY* genes act in the same non-linear DNA repair pathway [26]. The *uvsX* gene encodes a RecA-like recombinase [22,23,29], whereas *uvsW* and *uvsY* [24,26,27,29] encode a DNA helicase and a RMP, respectively. The SSB-like protein of T4 is called gp32 and plays a role in both DNA replication and repair similar to its bacterial homolog [30]. Genetic studies revealed an intricate link existing between UvsY and DNA replication. In the second stage of infection, HR is used as the main source of replication via the recombination dependent replication (RDR) mechanism involving UvsY [31]. 

The bacteriophage λ (lysogenic) also encodes its own recombination system called λRed able to repair ss- and dsDNA breaks. The λRed combines an exonuclease (5′ to 3′) Exo, a Gam protein, which prevents the action of the host RecBCD and a single-strand binding protein involved in DNA annealing, called Beta [32]. Additionally, λphage encodes an accessory RMP, called λOrf (or NinB) [12,13], dispensable to the λRed recombination mechanism. Due to its broad adaptation to its host, λOrf can substitute for RecFOR in the process of SSB displacement to facilitate RecA loading [12,13,33]. 

### 1.3. Discovery of the Eukaryotic RMP—RAD52/BRCA2

RAD52 was identified as a radiation-sensitive mutation in *Saccharomyces cerevisiae* [34]. Additional genetic studies revealed its implication in homologous recombination, working alongside other genes (Table 1), all belonging to the RAD52 epistatic group. Genes of this group encode several proteins organized into two subgroups: (1) members of the MRX complex, RAD50, MRE11, and XRS2 involved in the recognition, resection and protection of the DSB; (2) other proteins involved in HR, notably the recombinase RAD51 and RAD51 nucleofilament regulators (RAD54, RAD55, RAD57, RAD59 and TID1). Interestingly, a RAD52 deletion is 250-fold more sensitive to ionizing radiation compared to RAD51 [35]. This greater recombination defect is due to the loss of the second function of RAD52 in DNA repair, via its role in single-strand DNA annealing [36,37]. Although essential for HR in *S. cerevisiae*, the RAD52 deletion only presented a subtle HR deficient phenotype in vertebrates, suggesting the existence of another DNA repair mechanism in this phylum. Beyond this, recent work uncovered a new mitosis-specific functions of RAD52 in mammals, showing that RAD52 promotes a break-induced replication-like pathway, dubbed mitotic DNA synthesis, that allows completion of DNA replication during chronic stress [38,39,40]. This function appears to be crucial for maintaining of telomeres via recombination [41,42].

The breast cancer susceptibility gene *BRCA2* is another crucial gene for HR in eukaryotes and was first identified in humans [43]. Analysis of truncated BRCA2 mutants in mouse uncovered its interaction with RAD51 [44]. The interaction between BRCA2 and the SSB-like protein RPA was established using Hela cells [45]. Together, these observations classify BRCA2 as a RMP facilitating homologous recombination at double-strand breaks during S and G2 phases of the cell cycle, when sister chromatid is available. In *Ustilago maydis* and mammalian cells, BRCA2 also acts as RMP during meiosis by facilitating the loading of DMC1 (RecA-like) onto ssDNA [46,47]. RAD52 was later found to be synthetically lethal to BRCA2 in humans despite the minor phenotype initially observed for the single-deletion [48]. RMP conservation from one organism to another is varied. Yeast only encodes RAD52, whereas *Caenorhabditis elegans* only encodes BRC2, mammalian cells encode both and show a different extent in HR phenotypes for the single BRCA2 and double knockdown BRCA2/RAD52 [48,49]. 

## 2. Biochemical Properties and Structural Insights on RMPs

Biochemical and structural studies were critical to understand both common function and specificities of RMPs. Based on their overall structure, RMPs can be grossly classified into two groups, globular RMPs harboring oligonucleotides/oligosaccharides binding (OB) fold domains such as RecO or BRCA2, and RMPs adopting a multimeric ring-like shape such as UvsY or RAD52 [50]. The following chapter summarizes the information available on structure and biochemical properties of RMP orthologs.

### 2.1. Biochemical Properties and Structure of the Bacterial RMPs—RecFOR 

The entire RecO protein structure has been solved for *D. radiodurans* and *E. coli* either alone or in complex with RecR or SSB [51,52,53,54,55]. RecO is a globular protein composed of three domains: (1) a N-terminus domain (NTD) folded in an OB fold motif forming a positively charged groove conserved throughout bacteria and suitable for DNA binding, followed by (2) a zinc finger binding domain and (3) a C-terminus domain located in the center of RecO [51]. Bulk experiments conducted with purified RecO revealed its ability to bind ss- or ds-DNA and anneal complementary DNA molecules [51,56,57]. RecO point mutants further characterized the direct interaction of RecO with the C-terminal region of SSB and RecR [55,58]. 

The structure of the bacterial RecR has been solved for *D. radiodurans*, *Pseudomonas aeruginosa*, *Thermoanaerobacter tengcongensis*, and *Thermus thermophilus* [59,60,61,62]. RecR protein consists of two domains: (1) a N-terminus domain accommodating a cavity suitable for dsDNA binding and (2) a C-terminus domain containing a Topoisomerase/primase (TOPRIM) domain and a Walker B motif. Structural and bulk assays realized with various bacterial RecR proteins revealed the formation of either a dimer or a tetramer, the latter assembles a ring-shaped structure [59,62]. The ability of RecR in binding DNA varies between different species and is usually ATP independent [59,63,64]. RecR interacts with either RecO or RecF through the same acidic residue clusters of the RecR TOPRIM motif, size exclusion analysis demonstrated preferred binding to RecF [62]. It has been suggested that RecR forms tetramers upon interaction with binding partners [65]. Structural analysis of the RecOR complex uncovered an unexpected configuration in which two RecO monomers are bound at each side of the RecR tetramer [53]. With the exception of *B. subtilis*, the interaction between RecO and RecR is required to displace SSB tetramers from ssDNA [58,66]. When SSB-coated substrates are used, RecOR stimulates RecA activities: Strand invasion (D-loop formation), branch migration, ATPase activity, and TLS mutagenesis [11,58,67,68,69]. 

The structure of RecF protein was solved for *D. radiodurans* and *T. tengcongensis* (with and without ATP) [70]. RecF belongs to the ATP binding cassette (ABC) superfamily and is structurally similar to RAD50 head domain and structural maintenance chromosome ATPases (SMC) [70,71]. RecF protein binds ATP, ss- or dsDNA and only dsDNA binding is ATP dependent [72,73,74]. In the presence of ATP, RecF forms dimers on dsDNA. The weak RecF ATPase activity triggers the dimer dissociation from dsDNA via a conformational switch [71,74]. Impaired ATP hydrolysis as well as interaction with RecR were found to increase RecF dsDNA binding [64,74]. Although not presenting an increased affinity for gapped DNA, RecFR complexes randomly bind to dsDNA and block RecA filament extension along dsDNA. When present in concentrations sufficient to coat the dsDNA, RecFR complexes constrict RecA nucleofilament to adjacent ssDNA in gaps [64,75]. Opposed to this observation, it has been suggested that RecF preferentially binds at 5′ end using m13-DNA annealed with short primers [76]. Notably, in this study, cssDNA m13 with a short dsDNA region was used which primarily offers only a unique dsDNA region near ssDNA for RecF or RecFR to bind. Generally, on gapped DNA, RecF has a variety of observed effects on RecOR RMP function [11,68,69,77]. Additionally, RecF antagonizes the RecX destabilization effect of RecA* filaments through a direct RecF/RecX interaction [78]. Consistent with its RMP function in DSB repair in organisms lacking RecBCD, RecFOR has been successfully used to reconstitute the first step of DSB repair [79]. Importantly, in all biochemical studies, no complex including both RecO and RecF has ever been detected. RecR forms complexes with RecO, and alternatively with RecF but not with both proteins at the same time [62,68]

Overall, biochemical and structural studies agree on a consensus model in which RecOR cooperatively act as RMPs (Figure 1). However, the function of RecF remains unclear except that it appears to have some role in the metabolism of ssDNA gaps.

### 2.2. Biochemical Properties and Structure of the Bacteriophage RMPs—UvsY and λOrf

RMP UvsY in T4 adopts a heptameric open barrel conformation [80] which wraps ssDNA upon binding. UvsY further forms a complex with gp32-ssDNA in which gp32 and UvsY interact with a 1:1 stoichiometry [81,82]. UvsY is proposed to both (1) exchange gp32 with UvsX molecules via direct interaction, and (2) further destabilize gp32 linear filament structure by bending the DNA. Thus, UvsY stabilizes UvsX presynaptic filaments and stimulates the UvsX activities in HR and replication (Figure 1). 

Structurally, RMP λOrf in bacteriophage λ is composed of two domains, (1) a small N-terminal domain and (2) a larger C-terminal domain [33]. λOrf forms an asymmetric homodimer adopting a ring conformation with a funnel-like channel in the center and C-tails extending away. The positively charged central cavity is proposed to accommodate ssDNA binding, but is predicted to be too small for dsDNA binding [33]. In agreement, gel shift and stopped-flow assays showed that λOrf is not able to bind dsDNA [33]. In addition, a direct interaction with *E. coli* SSB was demonstrated by far western-blot [33]. Sequence analysis revealed high conservation of the λOrf through members of the bacteriophage λ family, suggestive of RMPs having a common role in phage and host recombination system [90]. 

### 2.3. Biochemical Properties and Structure of the Eukaryotic RMPs—RAD52 and BRCA2

The homologous pairing domain of RAD52 forms an undecamer ring folded into a mushroom-like structure consisting of a “stem” and a “domed cap” [83]. The stem region harbors a fold commonly found in DNA/RNA binding proteins. Point mutants and the structure of the RAD52-DNA complex were used to identify two DNA binding sites, termed inner and outer, which are involved in ss and ds-DNA binding, respectively [36,83,84,85]. The outer binding site of RAD52 is essential for ssDNA annealing activity, stimulation of Rad51 D-loop formation and DNA supercoiling [84,85]. RPA heterotrimer also stimulates RAD52 annealing activity [37,91]. Further, in strand exchange reactions, RAD52 can target RAD51 to ssDNA and help overcome the inhibitory effect of RPA in order to facilitate the RAD51 presynaptic filament formation (Figure 1, [37,86]). 

In the case of BRCA2, only the structures of truncated C-terminus (~800 residues) and short N-terminus have been solved. BRCA2 was solved in complex with its various binding partners DSS1, DSS1, and DNA, RAD51 or PALB2 [50,87,88,92]. The full sequence analysis of BRCA2 predicts a structure containing three regions: (1) an N-terminus region separated by (2) an intrinsically disordered loop harboring highly conserved BRC repeats (eight in human) from (3) its C-terminus [47]. The C-terminus region has five domains: An N-terminal helical domain, two OB fold domains (OB1 and OB2), a tower domain, and a third OB fold domain (OB3). The OB2, tower, and OB3 domains interact with ssDNA whereas the helical domain interacts with DSS1 [47]. DSS1 has been recently shown to (1) destabilize the multimeric state of BRCA2, usually found in solution [87], favoring its monomeric form involved in HR and (2) displace RPA from ssDNA by mimicking DNA through negatively charged residues [87,89]. Furthermore, structural and bulk experiments revealed a direct interaction between BRCA2 and the recombinases: RAD51 and DMC1. This interaction between BRCA2 and the recombinases enhances presynaptic filamentation as well as D-loop formation [47,88]. The N-terminal BRCA2 peptide interacts with the C-terminus of PALB2, possibly allowing PALB2 associated with BRCA2 to promote BRCA2 recruitment to double-stranded breaks that are assembled as subnuclear foci [92]. Within these DNA damage clusters, PALB2 together with BRCA2 also helps promote D-loop formation by stabilizing RAD51 filament formation and enhancing its recombinase activity [93]. Together, these observations suggest that BRCA2, after being targeted to the nucleus in complex with PALB2, interacts with DSS1 and ssDNA to form active monomers that are able to bind and load recombinases on ssDNA freed of RPA by DSS1 (Figure 1). 

## 3. Biophysical Tools to Capture Information of Dynamic Biochemical Reactions

Genetics, structural and bulk-biochemical studies have provided a wealth of knowledge about the role of RMPs. Nonetheless, these methods come with limitations and may have missed spatiotemporal information of the reaction dynamics. They may have overlooked transient or underrepresented intermediates in the reaction population, or even that biochemical reactions may proceed via different pathways. These limitations can however be addressed by employing biophysical tools. In this chapter, we discuss *in vivo* and *in vitro* biophysical tools used to capture population dynamics of biochemical reactions and their intermediates. 

### 3.1. In vivo Biophysical Tools 

*In vivo* biophysical tools provide spatiotemporal information of biochemical reactions. Utilizing microfluidic devices, cells are flattened on a coverslip in the imaging plane (Figure 2A). This setup allows capturing the cellular response, for instance, after the addition of DNA damaging compounds or fixative to halt cellular processes prior to imaging [94]. Further, using microfluidic devices allows constant flow of nutrients and oxygen. Flow cells with channels direct cells to grow in line [95] simplifying the use of segmenting and single-cell tracing tools (Figure 2A). 

To visualize a protein in living cells, a protein can be covalently linked with a fluorescent protein [96]. Their linkage is genetically introduced [95,97], conventionally at N- or C-terminal locus and occasionally in the middle of the gene sequence such as in the case of SSB-mTur2 [98]. It is worth mentioning that fluorescent tags can inhibit protein function and alter expression and degradation levels. Further, some fluorescent protein probes require tens of minutes to fold and fluoresce [99], leading to a delay for the observation window. 

In wide-field near-TIRF microscopy (Figure 2B, [100]), cameras capable of single photon sensitivity are used to record the dynamics of individual molecules in immobilized cells. Burst acquisitions recorded at video rate (30 frames/second) cannot resolve freely diffusing molecules (D ≈ 10 μm^2^/s, [101]), resulting in a blurred signal. Molecules that can be resolved as distinguishable particles at video rate diffuse much slower, i.e., DNA-bound molecules (D ≈ 10^−5^ μm^2^/s, [101]). These static foci can be resolved against a background of ~100 freely diffusing molecules over time, called detection by localization (Figure 2C, [96,101,102]). Burst acquisitions capturing the different diffusion modes of fluorescent protein fusions provide information to determine binding times of the tagged protein. On the other hand, time-lapse movies, i.e., single frame collection of the same cells every 10 min [21], can inform on the spatiotemporal behavior of a protein over a period of hours. The fluorescence intensity of single-cells can also be used to track expression levels over time [103,104], informative for designing *in vitro* experiments. 

Imaging immobilized living *E. coli* cells, we recently investigated the behavior of RMPs RecF and RecO [21], historically described to form an epistasis group [5,8,12,13,105,106,107,108,109,110]. In contrast to expectations, we found that RecF and RecO have distinct cellular localizations in response to DNA damage when recording two-colour time-lapse of *recF-YPet recO-mKate2*. *E. coli* cells produced ~18 RecF molecules and ~12 RecO molecules per cell, corresponding to ~5 nM and ~4 nM, respectively. Despite similar concentrations, RecF formed foci more frequently (2.2 ± 0.2 RecF-YPet and 0.3 ± 0.1 RecO-YPet foci per cell). Upon UV exposure, cellular intensities stayed constant, giving no increase in protein concentrations, the number of foci per cell however increased (60 min after UV: 6.1 ± 0.7 RecF-YPet and 0.5 ± 0.2 RecO-YPet foci per cell, Figure 2C for RecO). RecF and RecO rarely bind to the same binding site at the same time. RecF often colocalizes with a replisomal marker, showing even increased replisome colocalization upon UV exposure. Furthermore, in response to UV damage, RecF foci predominantly contained two RecF molecules, suggestive of RecF dimerization. Fewer RecF foci were found in *recR* deletion strains, indicating that RecR supports RecF binding to the nucleoid. In stark contrast to RecF behavior, RecO foci rarely colocalized with replisome markers and formed largely independently of RecR. Following UV exposure, RecO foci spatially redistributed to the region close to the cell membrane, sites where RecA* filaments reside [111,112,113]. These observations indicate that RecF and RecO have distinct functions in the DNA damage response. Increased RecF-replisome colocalization after UV exposure supports the hypothesis that RecF has some role related to DNA replication following DNA damage. Future studies are necessary to uncover the mechanism of *E. coli* RMPs in facilitating displacement of SSB displacement by RecA recombinase.

Eukaryotic RMP behavior after DNA damage induction has been investigated in two independent single-molecule imaging studies. The first study demonstrated that the percentage of cells containing Rad52-GFP foci increased by a factor of 20 in response to ionizing radiation [114]. Rad52 molecules are then frequently colocalized with Rad51 recombinase foci. Essers and coworkers further found that fewer fluorescent foci were observed than DSBs generated at a given dose of ionizing radiation, indicative of multiple DSBs being processed within fluorescent repair structures. Alternatively, multiple Rad52 homo-oligomers may be involved in the repair of a DNA lesion, thereby generating high-affinity sites for other recombination proteins, such as Rad51 recombinase. In the second study, RMP BRCA2 showed slowed diffusion in response to DNA damage, such as ionizing radiation, HU and MMS [115]. Employing fluorescence correlation spectroscopy in living mammalian cells, BRCA2 frequently exhibited transient binding and appeared to form multimeric clusters, estimated to contain two to five monomers. This multimeric behavior may have implications for BRCA2 spatiotemporal function in homologous recombination *in vivo* as only monomers appear to be the functional form *in vitro*. Binding time and percentage of bound molecules increased upon DNA damage induction with three molecules per cluster on average. BRCA2 concentration in cells was determined to be 3–15 nM. In comparison, Rad51 recombinase concentration is ~100 nM. Given the observation that BRCA2 and Rad51 display similar diffusion behavior, this suggests a stoichiometry of 6:1 for Rad51:BRCA2 complexes [115]. Chaperoning of Rad51 may control its polymerization, allowing filament nucleation only at target sites. Taken together, both studies reveal that RMPs bind at sites of recombinase loading. Spatiotemporal data suggest a complicated mechanism, possibly involving repair hotspots and an interplay of affinities.

### 3.2. In vitro Biophysical Tools 

*In vitro* biophysical tools have been frequently used to study the function of recombinase proteins and provide a quantitative description of protein-DNA interaction. Surface plasmon resonance (SPR) has been used to determine binding affinities by measuring kinetics of mass adsorption or desorption on a chip (Figure 3A). Single-molecule optical approaches have been employed to measure reaction dynamics in HR [116,117]. These approaches are based on readouts of mechanical properties of DNA upon protein binding or detection of fluorescent signal of labelled protein or DNA. Techniques involving mechanical manipulation are optical and mechanical tweezers (Figure 3B, [118,119]), techniques utilizing fluorescence signal are single-molecule imaging of fluorescent molecules and single-molecule FRET (Förster Resonance Energy Transfer [120,121], Figure 3C) spectroscopy. Biophysical approaches also frequently employ microfluidic flow chambers, allowing the change of buffer or addition of reaction components. Techniques to characterize structural and functional properties involve atomic force microscopy (AFM, see Figure 3D) which can achieve sub-nanometrer resolution, retrieving information such as size and shape of protein complexes [122]. Together *in vitro* biophysical tools have started to shed light on the function of RMPs, elucidating a conserved mechanism where RMPs form a ternary complex with single-stranded binding protein and ssDNA.

Gajewski and co-workers examined the formation of the ternary complex of bacteriophage T4 recombination mediator UvsY, single-stranded binding protein gp32 and ssDNA utilizing SPR (Figure 3A, [80]). C-terminal deletion mutant of gp32, gp32ΔC240-301, does not support formation of ternary complex, whereas the N-terminal deletion mutant, gp32ΔN1-21 can still form ternary complexes. Structural data on UvsY informed on point mutants with reduced binding to ssDNA, UvsY(K58A) and UvsY(F73A). UvsY(K58A) further showed reduced ternary complex formation with gp32-ssDNA.

Manfredi et al. visualized a bridged complex of *B. subtilis* RecO, SsbA and ssDNA using AFM [123]. Products of SsbA binding to single-stranded DNA (ssDNA) showed reduced secondary ssDNA structures. Addition of RecO to SsbA-ssDNA resulted in the formation of bridged protein complexes. RecO destabilizes SsbA-ssDNA complexes. At increased RecO concentration, RecO bridging SsbA-ssDNA structures promotes SsbA dislodging and strand annealing upon homology search.

A recent FRET study reported the ternary complex formation between *D. radiodurans* RecO, SSB and ssDNA [124]. A DNA template containing ssDNA overhang with acceptor and donor dye towards each end was used to monitor the conformational changes of 70 nt ssDNA. FRET efficiency of freely diffusing labelled DNA is 0.09, whereas efficiency increases in the presence of SSB (E = 0.58) and RecO (E = 0.78), suggestive of fluorescence donor and acceptor being in closer proximity to each other. Titration of either SSB or RecO further allowed measurements of their dissociation constants in complex with ssDNA. SSB shows a very slow off-rate compared to RecO (SSB K_D_ = 0.28 ± 0.01 nM, RecO K_D_ = 79 ± 11 nM). At equimolar ratios of RecO and SSB-ssDNA, the authors observed slow displacement of SSB by RecO. From 40 min after RecO addition, ~60–70% of DNA molecules displayed a FRET efficiency of 0.78, suggesting SSB displacement by RecO. DNA-bound RecO could however not be displaced by SSB. Upon immobilization of DNA on coverslips, using FRET traces, the authors found that RecO association with ssDNA is slowed in the presence of SSB (ssDNA 8.4 × 10^4^ M^−1^ s^−1^, ssDNA-SSB 7.5 × 10^1^ M^−1^ s^−1^). In the process of SSB displacement, traces showed a low FRET intermediate, indicating ssDNA extensions, potentially due to the formation of a RecO–ssDNA–SSB ternary complex. Interestingly, most displacements occurred after this low FRET intermediate state. The authors further discovered that RecO has two main binding states, one at a low FRET where DNA is extended and one at a high FRET where ssDNA ends are close, consistent with ssDNA wrapping around RecO. Mutation of positively charged N-(K35E/R39E) or C-terminal residues (R195E/R196E) impaired the formation of a high FRET state, suggesting that the first binding state involves stretching of ssDNA similarly seen during SSB displacement. These mutants showed a large reduction in SSB displacement and an increase in intermediate state. The authors proposed a sequential mechanism (Figure 3C): 1. Diffusion of SSB on ssDNA [125] may expose ssDNA for RecO binding 2. RecO binds to ssDNA-SSB to form a ternary complex 3. Positively charged residues of RecO allow the displacement of SSB from ssDNA despite the large difference in their binding affinity 4. RecO transitions to its second binding mode, wrapped in ssDNA.

Similarly, a single-molecule FRET spectroscopy study demonstrated that eukaryotic RMP Rad52 modulates the dynamics of RPA on ssDNA in *S. cerevisiae* [128]. RPA has six DNA-binding domains (DBDs) with DBD-A positioned closer to the 5′ end of ssDNA and DBD-F closer to 3′ end of ssDNA. FRET traces of DBD-A or DBD-D labelled with Cy5 on ssDNA labelled with Cy3 close to 5′ or 3′ end, respectively, fit a four-state mode, revealing conformational dynamics. In the presence of Rad52, state 4 of DBD-D is lost, consistent with Rad52 modulating RPA binding dynamics on ssDNA. Future studies will show if this ternary complex facilitates Rad51 recombinase loading.

Shivji et al. demonstrated that BRCA2 increases Rad51 recombinase assemblies on ssDNA [129]. Filament-like assemblies were also visualized by electron microscopy (Figure 3E). In agreement with another study [115], BRCA2 inhibits Rad51 assemblies on dsDNA, suggesting that BRCA2 chaperones Rad51 thereby only allowing filament formation at target site. In light of strand exchange reactions, further experiments showed that BRCA2 and Rad51 together promote the formation of joint DNA molecules when mixing dsDNA with complementary ssDNA. The authors propose that the difference in efficiencies of RAD51 assemblies on ssDNA and dsDNA in the presence of BRCA2 facilitate strand exchange reactions.

A recent single-molecule imaging study demonstrated that BRC-2 (BRCA2) and Rad51 paralogs, RFS/RIP-1, stimulate nematode Rad51 filament growth [130]. Further, Belan and co-workers showed that BRC-2 and RFS/RIP-1, synergistically, can even further increase the efficiency in Rad51 filament assembly. This observation is consistent with the notion that BRC-2 initially facilitates Rad51 nucleation and RFS/RIP-1 then supports nascent filament growth. Opposed to RecA filament growth directionality, RFS/RIP-1 promotes Rad51 filament growth in 3′ to 5′ direction. During this process, RFS/RIP-1 stabilizes filaments on ssDNA, thereby transiently interacting with the 5′ end of the filament. This transient interaction depends on ATP binding/hydrolysis. An ATPase impaired mutant, RFS-1(K56A/R), is capable of stabilizing Rad51 filaments but HR-inefficient *in vivo*. These observations support a model where RMPs sequentially act in order to nucleate and grow Rad51 filaments.

Taken together, these studies uncovered fundamental principles about the role of RMPs in displacing the single-stranded binding protein, recombinase loading and nucleofilament stabilization across organisms. Employing biophysical tools revealed that the formation of a ternary complex, RMP-single-stranded binding protein-ssDNA, appears to be a conserved mechanism. 

## 4. Biophysical Tools to Further Investigate the Role of RMPs

Biophysical tools are powerful techniques used to uncover fundamentals of biochemical reactions. Despite the knowledge already gained on RMP’s molecular mechanisms, some questions are still outstanding. In this last chapter, we discuss how future studies could address several open questions using biophysical tools in combination with bulk and *in vivo* assays.

In living bacteria, single-molecule fluorescence observations contradict the accepted model of RecF and RecO proteins working in close proximity during DNA repair [21,76,131]. Distinct binding sites of RecF and RecO can however be explained by interaction and/or regulation orchestrated by other proteins present in the physiological context. It would be particularly interesting to study labeled RMPs during DNA repair alongside with (1) PAmC-mCI developed by Ghodke et al. to probe RecA nucleofilament [132], (2) functional SSB-mTur2, tagged in its intrinsic disordered loop [98], (3) MuGam-gfp to label DSB [133], (4) HaloTag RecB protein [134], and (5) fluorescently labeled TLS polymerases DinB and UmuCD [135,136]. Such experiments could add knowledge about RMP function in the greater context of DNA repair. Furthermore, the literature might be missing interaction partners of RMPs. Just recently, DR1088 was identified to interact with *D. radiodurans* RecF with RecO-like biochemical properties [137]. Identification and characterization of molecular interactions in *in vivo* pull downs could in the future be helped by mass photometry [138]. New interaction partners could also be discovered using high throughput methods of single-cell imaging coupled with single-cell barcoding and DNA sequencing [139].

The current knowledge of bacterial as well as phagic RMPs combined with future protein designing and biophysical experiments could allow the development of new generations of therapeutics in the fight against multidrug resistant bacteria. Phage therapy has been used by some countries [140], paving the way for developing new generations of “super phage” therapeutics that interact with and inhibit the HR host system.

The development of a new generations of cancer therapeutics were initiated after the observation of *BRCA2*-deficient cells being more sensitive to DNA damage, initiating IR and cisplatin treatment. Eventually, based on the finding that BRCA2 acts in both HR and base excision repair, new therapeutics such as PARPi were developed [141]. Similarly, the discovery of Rad52 and its role in HR has opened another option to explore for treating *BRCA2*-deficient breast cancer cells [48,142]. Dissecting and characterizing the levels of complexity of eukaryotic DNA repair mechanisms will undoubtably open possibilities for future treatment options. Techniques, such as iPOND (isolation of protein on nascent DNA) coupled with mass spectrometry, already helped to discover novel proteins involved in maintaining genomic integrity. HMCES, for instance, which is conserved through organism, is a single-strand DNA repair protein that senses and targets abasic sites [143,144]. Combining iPOND with CRISPR and *in vivo* single-cell imaging could improve the general understanding of RMPs [145], paving the way for the development of new therapeutics.

## Figures and Tables

**Figure 1 biology-10-00288-f001:**
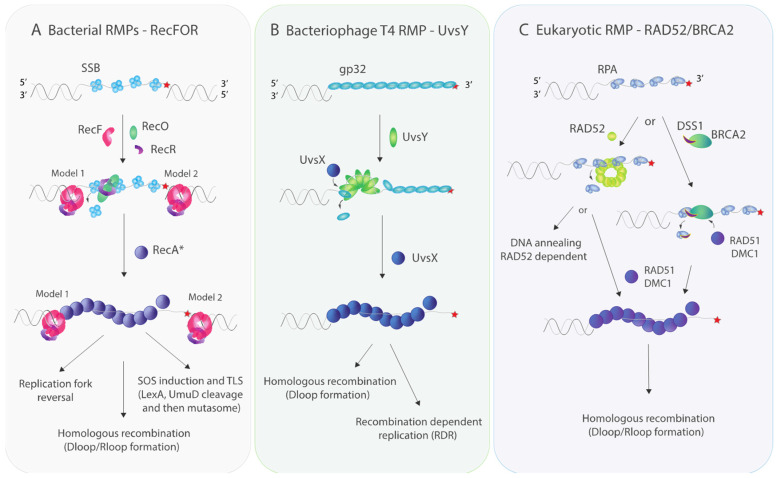
**Recombination mediators facilitate recombinase loading.** (**A**) Bacterial RMPs –RecF, RecO and RecR—facilitate loading of RecA recombinase. Two models have been proposed in the literature. Model 1 suggests that RecFR favors RecA nucleation at the 5′ end of the gap [76]. Model 2 proposes that RecFR bound to 3′ end limits RecA filament extension beyond the gap [64,75]. Both models describe RecO displacing SSB in the presence of RecR on ssDNA to facilitate RecA loading. In spite of a clear genetic relationship to RecO, the role of RecF is unclear. No complex containing both RecO and RecF has ever been detected and recent results lead to questions about the status of RecF as an RMP. (**B**) Bacteriophage T4 RMP—UvsY—adopts a multimeric ring-like shape [80]. Multimers of UvsY then destabilize gp32 filaments by bending the DNA and exchange gp32 with UvsX recombinase molecules via direct interaction, thereby stabilizing UvsX presynaptic filaments [80,81,82]. (**C**) Eukaryotic RMP—RAD52—forms undecamer rings [83] which can promote RAD52-depedent DNA annealing and RAD51 D-loop formation [84,85,86]. Eucaryotic RMP—BRCA2—interacts with DSS1 and ssDNA, whereby DSS1 breaks up multimers of BRCA2, forming active BRCA2 monomers on ssDNA [87,88,89]. These BRCA2 monomers then load RAD51 alone (mitosis) or both RAD51 and DCM1 (meiosis) onto ssDNA [47,88,89]. In the process of RAD51/DMC1 recombinase loading, DSS1 competes RPA off ssDNA either prior to or in the process of recombinase loading.

**Figure 2 biology-10-00288-f002:**
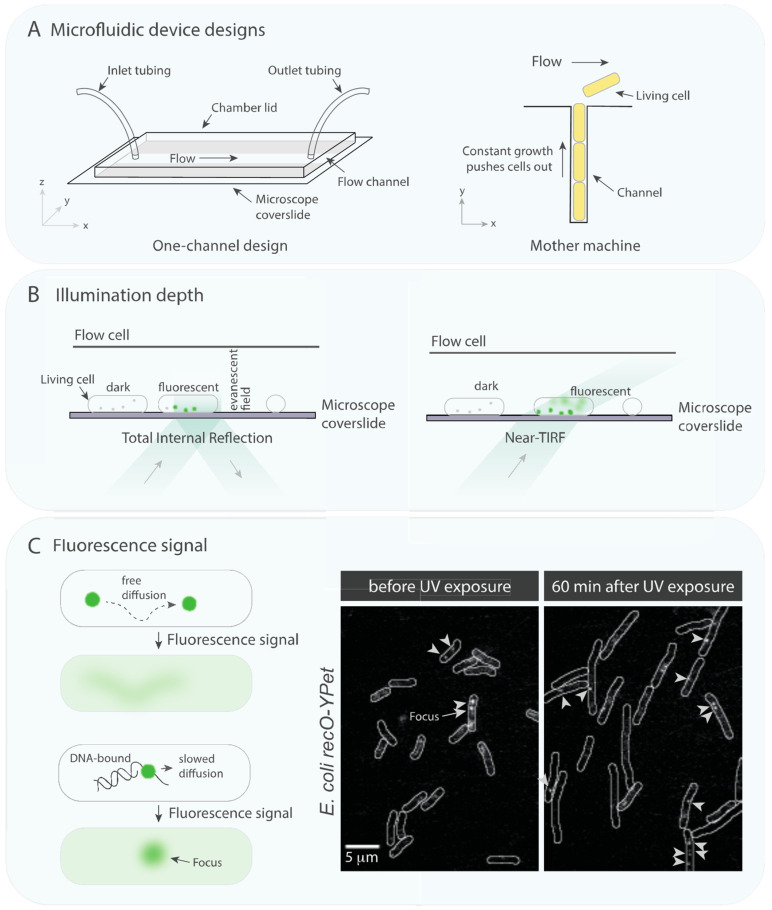
***In vivo* single-molecule fluorescence tools.** (**A**) Two designs of microfluidic devices to hold cells for live-cell imaging. **Left**: Simple single-flow channel device where cells can be immobilized on derivatized surface using, i.e., (3-aminopropyl)triethoxysilane. **Right**: Mother machine design [95] where cells are captured in a channel orthogonal to the flow channel, allowing to follow a single-cell during its cell cycle or in response to exogenous DNA damage. (**B**) **Left**: Total internal reflection fluorescence (TIRF) microscopy excites fluorophores in the evanescent field, close to the coverslip. **Right**: Near-TIRF setting allows excitation deeper into the cells, additionally capturing fluorescently tagged proteins further away from the coverslip, making it more suitable to follow proteins of interest in living cells. (**C**) Proteins that are not bound to DNA diffuse faster than proteins that are bound to the larger genome. The difference in diffusion behavior can be captured when imaging fluorescently tagged proteins in living cells. At video rate, the fluorescence signal of a freely diffusing protein appears as a blur, whereas the signal of DNA-bound proteins shows static foci. The figure shows exemplary fluorescence datcells before and 60 min after UV irradiation [21]. Some foci are pointed out with white arrows.

**Figure 3 biology-10-00288-f003:**
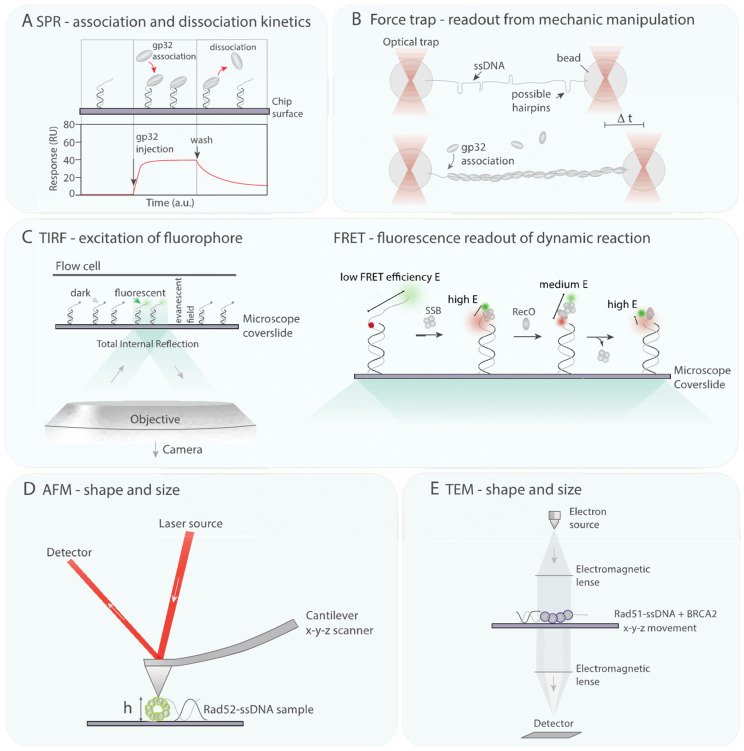
***In vitro* biophysical tools.** (**A**) Surface plasmon resonance (SPR) binding analysis is used to determine binding affinities. On a chip, measuring mass adsorption and/or desorption can unveal kinetics of binding processes. Illustrated here is association and dissociation of gp32 with ssDNA [80]. (**B**) Force traps (optical and mechanical tweezers) can hold each end of linear DNA, to determine differences in mechanical properties of DNA upon protein binding. Association of single-strand binding protein gp32 for instance prevents the formation of ssDNA secondary structure [126]. (**C**) **Left**: TIRF microscopy excites fluorescent molecules close to a microscope coverslip in evanescent field. Molecules can be bound to DNA that is tethered to the coverslip surface. **Right**: Förster Resonance Energy Transfer (FRET) spectroscopy coupled with TIRF setting permits to study molecular mechanisms of two molecules approaching at a distance of several nanometers. FRET applied to optical microscopy has been used to study the displacement of SSB by RecO on ssDNA, capturing a ternary intermediate state [124]. (**D**) Atomic Force Microscopy (AFM) is used to characterize structural and functional properties of intermediates and products of biochemical reactions. AFM tip scans an immobilized sample, x-y-z movements translate into a topograph. Rad52 binding to ssDNA has been shown to extend and unfold ssDNA [127]. (**E**) Transmission Electron Microscopy (TEM) is used to visualize proteins and protein-DNA complexes, generating a high-magnification image with a particle beam of electrons combined with TEM optics. Negative staining of samples of RAD51, ssDNA in combination with BRCA2 or omitting BRCA2 revealed that BRCA2 promotes RAD51 filament formation [125].

**Table 1 biology-10-00288-t001:** Overview of recombination mediators and partners in different organisms. This table lists RMPs, their interaction partners and functions in different organisms.

	1. Bacteria	2. Phages T4 or Lambda		3. Eukaryotes	
Single-strand binding protein	SSB	T4 gp32	*E.coli* SSB	RPA	
RMP(s)	RecF, RecO, RecR	UvsY	λOrf (NinB)	RAD52	BRCA2
Recombinase	RecA	UvsX	λBeta or *E.coli* RecA	RAD51/DMC1	Rad51/DMC1
Pathways facilitated by RMPs	HR, SOS, TLS, replication	HR and RDR	HR, SOS, TLS, replication, phage cycle	HR, SSDA	HR
Additional partners	RecQ, RecJ, RecN, RecG, RecX	UvsW helicase	λExo and λMu	RAD50, MRE11, XRS2, RAD54, RAD55, RAD57, RAD59, TID1	RAD50, MRE11, XRS2, PALB2, DSS1, BRCA1

## Data Availability

*E. coli* fluorescent fusion construct images were kindly provided by authors of Henrikus et al. 2019.

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
