# Peer review of "Elucidating Recombination Mediator Function Using Biophysical Tools"

_biology, 2021, doi:10.3390/biology10040288_

Round 1

Reviewer 1 Report

This is a well-written review article on the biology and biophysics of homologous recombination mediators (RMPs). It presents some historical genetics experiments that led to the discovery of RMPs and discusses contemporary biochemical data and biophysical methods that are used to perform structure-function studies on these essential proteins. Care is taken to present functional and structural similarities and divergences between RMPs from phages, bacteria, yeast and vertebrates making it quite interesting for a broad audience. Provided that a few minor points are addressed and a few typos corrected, this will be a fine addition to Biology journal.

Minor points :

  1. When discussing RMPs from vertebrates it would make more sense to state that BRCA2 functions during the S and G2 phases of the cell cycle to promote homologous recombination at double-stranded breaks instead of in mitosis (lines 112 and 193). In these organisms, HR is constrained in these 2 phases of the cell cycle and uses the sister chromatid as a template for break repair whereas non-homologous end joining functions mainly in G1 and acts as a minor repair pathway in the S and G2.
  2. In mammals, the functions of RAD52 during homologous recombination repair of double-stranded DNA breaks are limited, yet the protein is conserved. Recently, new mitosis-specific functions of RAD52 in mammals were discovered when it was shown that RAD52 promotes a break-induced replication-like pathway dubbed Mitotic DNA Synthesis that allows completion of DNA replication during chronic stress (Bhowmick et al. Mol Cell 2016, Minocherhomji et al. Nature 2015, Sotoriou et al. Mol Cell 2016). This function appears to be particularly important for the maintenance of telomeres via recombination (Min et al. Mol Cell Biol 2017, Verma et al. Genes and Development 2019). It would be timely to discuss these new functions of RAD52 in section 1.3 of the manuscript.
  3. Line 143. The sentence is unclear, the ability of RecR in binding DNA varies (between what, homologs from different bacterial species ?). This should be corrected.
  4. Lines 240-245. PALB2’s main function is not to bring BRCA2 to the nucleus. It would be more accurate to state that PALB2 associates with BRCA2 to promote its recruitment to double-stranded breaks that are assembled as subnuclear foci. There, PALB2 together with BRCA2 also helps promote D-loop formation by stabilizing RAD51 filament formation and enhancing its recombinase activity (reviewed in Ducy et al. TIBS 2019).
  5. I would remove or modify the sentence in lines 484-485 as cancer therapeutics predates the isolation of BRCA2 deficient cancer cell lines by quite a few years…
  6. The figures are rather small making it difficult to make out tiny details. It would be a good idea to increase their size.

Typos and syntax:

Line 19 : relative to the

Line 46 : were the first to use

Line 48 : involved in homologous recombination (HR), including recF

Line 147 : upon interaction (single space)

Line 172 : RecFOR has been

Lines 198-199 : which wraps ssDNA upon binding. UvsY further forms…

Line 224 : overcome the inhibitory

Line 232 : and a third

Line 284 In vivo

Line 287 in a channel orthogonal to the flow channel

Line 329 In the second study

Line 337 Was determined to be 3-15 nM

Line 341 at target sites.

Line 358 which can achieve sub-nanometer resolution

Line 369 close to a microscope coverslip

Line 374 intermediate state [122].

Line 405 displayed a FRET efficiency

Line 423 that eukary- (single space)

Line 426 traces of DBD-A

Line 458 powerful techniques

Reviewer 2 Report

Reviewer’s comments:

Full Title:  Elucidating Recombination Mediator Function Using Biophysical tools.

Summary of the main results of the article:

In “Elucidating Recombination Mediator Function Using Biophysical tools,” the authors present a review on the topic of Recombination mediator proteins in different domains of life. This topic is highly relevant to the scientific community. The review offers complete and neutral coverage of the subject of ribosomal elongation. The review is scientifically accurate, and the authors make an effort to unbiasedly select references that represent the topic, including current and past actors in the field. Specifically, the authors divide the document into four main sections: i) Discovery and phenotype of bacterial RMPs. ii) Biochemical properties and structural insights on RMPs. iii) Biophysical tools to capture information of dynamic biochemical reactions. iv) Biophysical tools to further investigate the role of RMPs.

  • Major comments:

Overall, I consider that the article is well-structured and well-written. I only noticed some minor errors in the grammar that need to be addressed before publication.

Minor comments.

The text lacks articles in multiple paragraphs. For example, line 415, currently reads as “[…] impaired the formation of high FRET state …” and it should be “[…] impaired the formation of a high FRET state …” or in Line 184, “[…] in presence of […]”   should read “[…] in the presence of […]”.  This problem is consistent in the text. Please double-check the complete document to make sure no errors like this persist on the revision.

The text also has the problem of not using hyphens on two-word adjectives before a noun. For example:

“single strand binding proteins” change to “single-strand binding proteins”

“single cell level” change to “single-cell level”

“radiation sensitive mutation” change to “radiation-sensitive mutation”

“single strand DNA annealing” change to “single-strand DNA annealing”

“single deletion mutants” change to “single-deletion mutants”

“stopped flow assays” change to “stopped-flow assays”

“high affinity sites” change to “high-affinity sites”.

Problems with commas and periods.

A common problem in the text is the lack of commas, for example, in a list you must follow the format “item 1, item 2, and item 3”. Check line 77 “Sedimentation, phage plaque and burst size […]” it should be “Sedimentation, phage plaque, and burst size[…]”.

Error on verbs:

Line 324 “Rad52 molecules then frequently colocalized with Rad recombinase foci […]” change to “Rad52 molecules are then frequently colocalized with Rad recombinase foci […]”

Line 356, “Techniques to characterize structural and functional properties involve atomic force microscopy (AFM,  see Figure D) which can achieved sub-nanometer resolution […]” change to “Techniques to characterize structural and functional properties involve atomic force microscopy (AFM,  see Figure D) which can achieve sub-nanometer resolution  […]”

Minor typos:

Line 33 “the crucial role of biophysical tools to investigate molecular mechanism” change to “the crucial role of biophysical tools to investigate molecular mechanisms”

Line 274 “i.e DNA-bound molecules” change to “i.e., DNA-bound molecules”

In Line 389 “et. al” change to “et al.” Specifically, in “Manfredi et. al” change to “Manfredi et al.”

Line 420.  “positively charged residues of RecO” Use upper case “Positively charged residues of”

Line 426. “FRET traces of of DBD-A or DBD-D” remove “of of ”

Problems with clarity in text.

Line 273. “Molecules that can be resolved at video rate diffuse much slower” this sentence is not very clear.
